# FISHER-AWARE QUANTIZATION FOR DETR DETECTORS WITH CRITICAL-CATEGORY OBJECTIVES

## ABSTRACT

The impact of quantization on the overall performance of deep learning models is a well-studied problem. However, understanding and overcoming its effects on a more fine-grained level is still lacking, especially for harder tasks such as object detection with both classification and regression objectives. This work identifies the performance for a subset of task-critical categories, i.e. the critical-category performance, as a crucial yet largely overlooked fine-grained objective for detection tasks. We analyze the impact of quantization at the category-level granularity, and propose methods to improve performance for the critical categories. Specifically, we find that certain critical categories have a higher sensitivity to quantization, and have inferior generalization after quantization-aware training (QAT). To explain this, we provide theoretical and empirical links between their performance gaps and the corresponding loss landscapes with the Fisher information framework. Using this evidence, we propose a Fisher-aware mixed-precision quantization scheme, and a Fisher-trace regularization for the QAT on the critical-category loss landscape. The proposed methods improve critical-category performance metrics of the quantized transformer-based DETR detectors. When compared to the conventional quantization objective, our Fisher-aware quantization scheme shows up to 0.9% mAP increase on COCO dataset. A further 0.5% mAP improvement is achieved for selected critical categories with the proposed Fisher-trace regularization.

## 1 INTRODUCTION

Object detection is a challenging core application in computer vision, which is crucial for practical tasks such as autonomous driving etc. Recent DEtection TRansformer (DETR) model (Carion et al., 2020) and its variants achieve state-of-the-art results on multiple detection benchmarks (Liu et al., 2022). However, their performance comes at the cost of large model sizes and complicated architecture. Therefore, quantization (Choi et al., 2018; Dong et al., 2020; 2019; Polino et al., 2018; Yang et al., 2021) is typically applied in real-world applications to reduce the memory footprint and inference latency time on cloud and edge devices (Horowitz, 2014). Inevitably, the perturbation of weights and activations introduced by the quantization process degrades the performance of floating-point models. Previous research mainly focuses on a *trade-off between the model size and the overall performance* (e.g. average accuracy for classification and mean average precision (mAP) for detection) of the quantized models (Dong et al., 2019; Yang et al., 2021; Xiao et al., 2022).

However, a *fine-grained performance objectives* are often more important than the overall performance in the real world (Barocas et al., 2019; Tran et al., 2022). Suppose an autonomous vehicle is processing a scene containing people, vehicles, trees, light poles, and buildings, as illustrated in Fig. 1 (left)[1]. Some non-critical objects (light poles, trees, and buildings) only need to be localized to avoid collision, yet misclassification within this group of categories is not as critical if they are all considered as "other obstacles". On the other hand, *critical classes* such as a person or vehicle require both accurate classification and localization for safe operation. The overall performance cannot distinguish between an error within non-critical categories vs. a critical object error. So, it is missing the granularity to represent the true task-critical objectives of real-world applications. Yet to the best of our knowledge, for both post-training quantization (PTQ) and quantization-aware training (QAT), the analysis of the impact on such task-critical fine-grained objectives of object detection models is overlooked.

---

[1]Street scene photo in Fig. 1 credits to Google Street View.

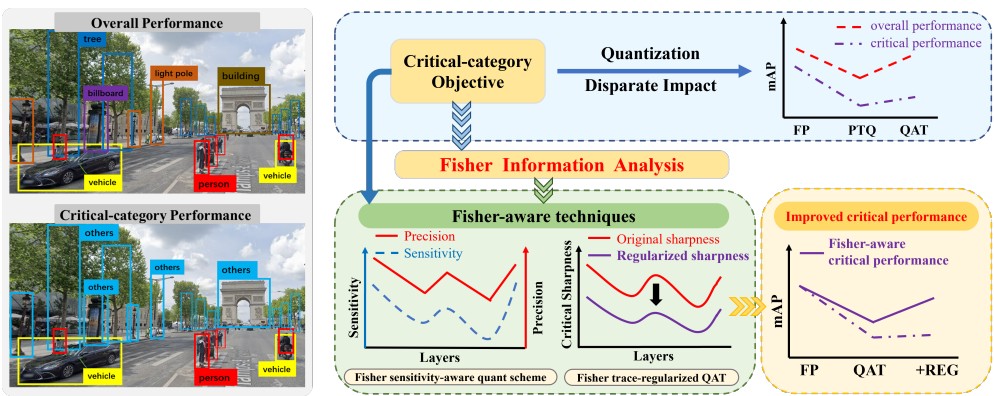

Figure 1: **Overview.** We investigate a practical setting with task-dependent critical-category objectives, which is formulated with label transformation in Sec. 3.2. We empirically observe disparate effects of quantization on the critical-category performance in Sec. 3.3, where PTQ and QAT lead to performance gaps for critical categories. With the theoretical analysis using Fisher information, we propose Fisher-aware quantization scheme and regularization in Sec. 4 to improve critical performance for quantized DETR models as demonstrated in Sec. 5 experiments.

In this paper, we follow this practical yet overlooked setting to formulate a set of task-critical objectives for DETR-based object detection models, accomplish a fine-grained quantization impact analysis, and propose techniques to improve the corresponding objectives. Specifically, we disentangle classification and localization objectives to define a fine-grained *critical-category performance* with non-critical label transformation, as shown in the updated bounding boxes in Fig. 1 (left). With this formulation, we introduce a thorough analysis of the impact of quantization on the critical-category performance of DETR model. As illustrated in Fig. 1 (right), we find that quantization has a disparate effect on the category-wise performance, where some groups of classes are more sensitive to quantization with up to 1.7% additional mAP drop. Furthermore, unlike the almost-guaranteed improvement of the overall performance after QAT, such training can increase performance gaps for some task-critical categories. We provide both theoretical and empirical analysis of such quantization effects using the loss surface landscape of the critical objectives using the Fisher information framework (Perronnin & Dance, 2007).

Based on this analysis, we propose two novel techniques: Fisher-aware mixed-precision quantization scheme, and Fisher-trace regularization. Both techniques optimize the landscape of critical objectives and, therefore, improve critical-category performance. Our experiments show consistent critical-category performance improvements for DETR object detectors with different backbones and architecture variants. The contributions of this paper are summarized as follows:

- We formulate the critical-category performance for object detection applications, and observe disparate effects of quantization on the performance of task-critical categories.
- We provide analytical explanations of the quantization effects on critical-category performance for DETR-based models using a theoretical link to the Fisher information matrix.
- We propose Fisher-aware mixed-precision quantization scheme that considers the sensitivity of critical-category objectives and improves corresponding detection metrics.
- We propose Fisher-trace regularization for the loss landscape of our objectives during quantization-aware training to further improve critical-category results.

## 2 RELATED WORK

**Object detection.** Object detection is a core task for visual scene understanding. Conventional object detectors rely on a bounding box proposals (Girshick, 2015), fixed-grid anchors (Redmon et al., 2016) or window centers (Tian et al., 2019). However, the performance of these methods is largely affected by the bounding box priors and the post-processing steps (Carion et al., 2020). The transformer-based DETR (Carion et al., 2020) provides a fully end-to-end detection pipeline without a surrogate tasks. Follow-up research further enhances DETR by introducing a deformable attention (Zhu et al., 2021),

query denoising (Li et al., 2022), and learnable dynamic anchors as queries (Liu et al., 2022). With the growing popularity of DETR-based architecture, we believe that understanding of quantization impact on DETR performance is an important topic, especially at the fine-grained level. Common object detection benchmarks evaluate fine-grained performance metrics that depend on the object size (Lin et al., 2014) or occlusion status (Geiger et al., 2012). However in practical applications, object type i.e. its category is often more important than the object size. This motivates us to further investigate detectors with critical-category objectives.

**Efficiency-performance tradeoff.** Multiple methods have been proposed to compress deep neural network (DNN) models, including pruning (Han et al., 2015; Wen et al., 2016; Yang et al., 2020b; 2023), quantization (Polino et al., 2018; Dong et al., 2020; Yang et al., 2021; Guo et al., 2022), factorization (Wen et al., 2017; Ding et al., 2019; Yang et al., 2020a), and neural architecture search (Wu et al., 2019; Cai et al., 2020) etc. This work explores the impact of quantization that is widely supported by the hardware (Horowitz, 2014) and can be almost universally applied to DNN compression in architecture-agnostic fashion. Post-compression model performance is the key focus in previous research. However, the overall i.e. average performance hides the important fine-grained metrics e.g. the results for certain groups of categories. Recent works (Tran et al., 2022; Good et al., 2022) analyze the disparate impact of pruning on classification accuracy, which leads to the fairness concerns (Barocas et al., 2019). Our work extends this direction and investigates quantization effects of DETR-based object detection at the critical-category performance granularity.

**Second-order information in deep learning.** Unlike conventional optimization with the first-order gradients, recent research has found the importance of utilizing second-order information to increase the generalization and robustness of DNN models. Sharpness-aware minimization (Foret et al., 2021) links a loss landscape sharpness with the model ability to generalize. The latter can be improved using a regularized loss with the Hessian eigenvalues (Yang et al., 2022) computed w.r.t. parameter vector. Hessian eigenvalues are also used as importance estimates to guide the precision selection in mixed-precision quantization (Dong et al., 2019; 2020; Yao et al., 2021). Given the difficulty of exact Hessian computation, Fisher information matrix is proposed as an approximation of the importance in pruning (Kwon et al., 2022). In this work, we link the disparate impact of quantization to the critical objectives with the second-order Fisher information, and propose Fisher-aware quantization and regularization methods to overcome the quantization effects on critical categories in object detection.

## 3 CRITICAL-CATEGORY PERFORMANCE ANALYSIS

In this section, we provide an introduction to the notations and the object detection training objectives for DETR model in Sec. 3.1; formulate our critical-category performance in Sec. 3.2; and finally empirically analyze the impact of quantization on such critical performance in Sec. 3.3.

### 3.1 PRELIMINARY

We recap the computation process of DETR-based object detectors, and provide relevant notations for the rest of this paper. We mainly focus our discussion on the output of DETR model rather than its architecture. Then, the following notation is applicable to both the original DETR (Carion et al., 2020) and its more advanced variants such as DAB-DETR (Liu et al., 2022), Deformable DETR (Zhu et al., 2021) as well as any other detector with the end-to-end architecture.

Given an input image $x$, the DETR-type model $f(\cdot)$ outputs a fixed-size set of $N$ bounding box predictions $f(x) = \{[\hat{p}_i, \hat{b}_i]\}_{i=1...N}$. In each bounding box prediction, $\hat{p}_i$ is the vector of classification logits and $\hat{b}_i$ is the vector of bounding box coordinates. The former $\hat{p}_i \in \mathbb{R}^{n+1}$ contains logits for $n$ classes and an empty-box class ($\varnothing$). The predicted bounding box $\hat{b}_i \in \mathbb{R}^4$ consists of 4 scalars that define the center coordinates as well as the height and the width relative to the image size.

During the training, annotation is provided for each training image as a set of ground truth boxes $y_i = \{[c_i, b_i]\}$, where $c_i$ is the target class label and $b_i$ defines the bounding box. A Hungarian matching process is performed to find the closest one-to-one matching between ground truth boxes and predicted boxes including those with "no object" $\varnothing$ predictions. The training loss is computed between each pair of matched boxes, which is defined as a linear combination of a classification loss $\mathcal{L}_{cls}(\hat{p}_i, c_i)$ for all boxes, and a box loss $\mathcal{L}_{box}(\hat{b}_i, b_i)$ for all non-empty boxes.

## 3.2 FORMULATION OF CRITICAL-CATEGORY OBJECTIVES

As discussed in Sec. 1, the overall performance metric evaluated on the validation dataset can be not the most effective objective in some real-world scenarios. Category-level fine-grained performance for some specific task-critical categories can be more crucial than the averaged metrics. Here we provide a practical definition of the critical-category objectives for the detection task, and a corresponding evaluation method when applied to DETR-type detectors.

In classification tasks, class-level performance is often defined as the accuracy or loss of the model on a subset of the validation dataset that contains objects belonging to a certain group of classes (Tran et al., 2022). However, such definition is not practical for object detection task, as each input image in the dataset contains multiple objects from different categories. Instead, this work defines the critical objective based on the entire validation dataset, but with a *transformed model outputs* and annotations during the loss computation in order to focus it towards a certain group of critical object categories.

Formally, suppose there are in total $n$ categories in the dataset. Without loss of generality, suppose the first $m$ categories are the critical ones for a certain task that requires both an accurate classification and localization. Then, the rest of categories ($m + 1$ to $n$) are non-critical and a misclassification between them is acceptable. This can be expressed by the transformed prediction $\hat{p}_i' \in \mathbb{R}^{m+2}$ as

$$\hat{p}_i'[j] = \begin{cases} \hat{p}_i[j] & j = 1 \dots m \\ \max \hat{p}_i[m+1:n] & j = m+1 \\ \hat{p}_i[n+1] & j = m+2 \end{cases}. \tag{1}$$

The $(m + 1)$-th category in $\hat{p}_i'$ corresponds to "others", which represents non-critical categories. The max function is used to avoid a distinction when classifying non-critical categories. The $(m + 2)$-th category in $\hat{p}_i'$ is used for $\varnothing$ class, which is originally defined as the $(n + 1)$-th category in $\hat{p}_i$.

The same transformation is also applied to the ground truth box label $c_i$, where all $c_i \in \{m+1, \dots, n\}$ are set as the $(m + 1)$-th label in the transformed $c_i'$. No change is applied to the bounding box coordinates for the predicted and ground truth bounding boxes as we only define critical performance at the classification granularity to have simplified yet practical and instructive setting.

Note that the logit transformation can be applied directly to the output of a trained DETR model without any change to its architecture or weights $W$. With both $\hat{p}_i'$ and $c_i'$ being transformed, the Hungarian matching, loss computation, and mAP computation can be performed without modification. We name the loss computed with the original $\hat{p}_i$ and $c_i$ as *"Overall objective"* and it is expressed as

$$\mathcal{L}_A(W) = \sum_{i=1}^N \left[ \mathcal{L}_{cls}(\hat{p}_i(W), c_i) + \mathcal{L}_{box}(\hat{b}_i(W), b_i) \right]. \tag{2}$$

Similarly, the *"Critical objective"* is defined with the transformed $\hat{p}_i'$ and $c_i'$ as

$$\mathcal{L}_F(W) = \sum_{i=1}^N \left[ \mathcal{L}_{cls}(\hat{p}_i'(W), c_i') + \mathcal{L}_{box}(\hat{b}_i(W), b_i) \right]. \tag{3}$$

Each objective corresponds to either the *"Overall performance"* or the *"Critical performance"* when evaluating the mAP detection metric with the original or transformed outputs and labels, respectively.

## 3.3 EMPIRICAL EVIDENCE OF POST-QUANTIZATION GAPS

With the defined objective, we analyze how quantization affects the critical performance of DETR model. Specifically, we start with the official pretrained checkpoint of DETR with ResNet-50 backbone[2]. We use a symmetric linear quantizer $Q(\cdot)$ (Dong et al., 2019) to quantize a weight tensor $W$ to $q$ bits, which can be expressed by

$$Q(W) = \text{Round} \left[ W \frac{2^{q-1} - 1}{\max(|W|)} \right] \frac{\max(|W|)}{2^{q-1} - 1}. \tag{4}$$

---

[2]https://dl.fbaipublicfiles.com/detr/detr-r50-e632da11.pth

Table 1: Critical mAP for each super category before and after 4-bit uniform quantization.

| Super category | Person | Vehicle | Outdoor | Animal | Accessory | Sports | Kitchen | Food | Furniture | Electronic | Appliance | Indoor | Overall |
|---|---|---|---|---|---|---|---|---|---|---|---|---|---|
| Pretrained | 39.4 | 43.9 | 44.4 | 42.5 | 44.6 | 44.2 | 44.8 | 44.7 | 44.7 | 43.8 | 43.9 | 44.9 | 41.9 |
| PTQ 4-bit | 20.1 | 23.3 | 23.9 | 22.3 | 23.9 | 23.7 | 24.2 | 23.9 | 23.7 | 23.4 | 23.5 | 24.0 | 20.9 |
| mAP drop | 19.3 | 20.6 | 20.5 | 20.2 | 20.7 | 20.5 | 20.6 | 20.8 | 21.0 | 20.4 | 20.4 | 20.9 | 21.0 |

Table 2: Critical mAP for each super category of 4-bit quantized model before and after QAT.

| Super category | Person | Vehicle | Outdoor | Animal | Accessory | Sports | Kitchen | Food | Furniture | Electronic | Appliance | Indoor | Overall |
|---|---|---|---|---|---|---|---|---|---|---|---|---|---|
| PTQ 4-bit | 20.1 | 23.3 | 23.9 | 22.3 | 23.9 | 23.7 | 24.2 | 23.9 | 23.7 | 23.4 | 23.5 | 24.0 | 20.9 |
| QAT 4-bit | 34.6 | 38.6 | 39.2 | 37.2 | 39.4 | 38.9 | 39.5 | 39.3 | 39.2 | 38.6 | 38.6 | 39.6 | 36.7 |
| mAP gain | 14.5 | 15.3 | 15.3 | 14.9 | 15.5 | 15.2 | 15.3 | 15.4 | 15.5 | 15.2 | 15.1 | 15.6 | 15.8 |

We quantize all trainable weights in the DETR model with an exception of the final feed-forward (FFN) layers for the class and bounding box outputs. Quantization of these FFN layers leads to a catastrophic performance drop in the PTQ setting (Yuan et al., 2022). A 4-bit quantization is applied uniformly to the weights of all layers for all experiments in this section.

For critical performance, we define critical categories based on the "super category" labels in the COCO dataset (Lin et al., 2014). In total, 12 super categories are available in the COCO, where each contains from 1 to 10 categories of similar objects. For each selected super category, we consider all the categories within it as critical categories, while the rest of categories as non-critical and transform their logits and labels accordingly. The mAP measured at the transformed output is denoted as the critical mAP of this super category. For example, when measuring the critical performance of "Indoor" super category, "book", "clock", "vase", "scissors", "teddy bear", "hair drier", and "toothbrush" are considered as critical categories (the first $m$ categories in the Eq. (1) logit-label transformation), while others are set as non-critical. We perform such evaluation for all 12 super categories to understand the category-level impact of quantization on DETR.

As shown in Tab. 1, quantization has a disparate impact on the critical performance of the DETR model. The mAP drop after quantization has an up to 1.7% gap. We further perform 50 epochs of QAT and report the critical performance in Tab. 2. The performance increases differently for each super category with a gap of up to 1.1% mAP.

## 4 METHODS TO OVERCOME CRITICAL-CATEGORY QUANTIZATION EFFECTS

In this section, we provide theoretical analysis on the cause of empirical performance gaps in Sec. 4.1. Then, we propose our methods to improve such performance from the aspect of quantization scheme design and quantization-aware training objective in Sec. 4.2 and Sec. 4.3, respectively.

### 4.1 CAUSE OF POST-QUANTIZATION PERFORMANCE GAPS

For a pretrained DETR model with weights $W$, we investigate how quantization affects the critical objective $\mathcal{L}_F(W)$. We obtain the following two insights:

**Insight 1: Larger Fisher trace of critical objectives results in a higher sensitivity to weight perturbation.** The quantization process replaces the floating-point weights $W$ of the pretrained DETR model with the quantized values $Q(W)$ as in Eq. (4). Effectively, this perturbs the weights away from their optimal values, which leads to an increase in the critical objective value. With the second-order Taylor expansion around $W$, the quantization-perturbed loss $\mathcal{L}_F(Q(W))$ can be approximated using the non-perturbed objective $\mathcal{L}_F(W)$ as

$$\mathcal{L}_F(Q(W)) \approx \mathcal{L}_F(W) + g^T \Delta + \frac{1}{2} \Delta^T H \Delta, \qquad (5)$$

where the gradient $g = \partial \mathcal{L}_F(W) / \partial W$, the Hessian $H = \partial^2 \mathcal{L}_F(W) / \partial W^2$, and the weight perturbation or, in other words, the quantization error $\Delta = Q(W) - W$.

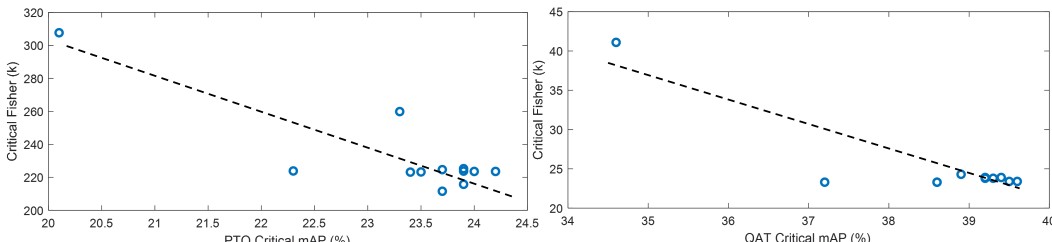

Figure 2: **Fisher trace of the critical-category objective vs. critical mAPs.** Performance is evaluated for the 4-bit quantized model after PTQ (left) and QAT (right), respectively.

We assume that the pretrained model converges to a local minimum. Then, the first-order term can be ignored because $g$ is close to zero (LeCun et al., 1989). Hence, Eq. (5) can be rewritten as

$$\mathcal{L}_F(Q(W)) - \mathcal{L}_F(W) \propto \Delta^T H \Delta. \tag{6}$$

For large models such as DETR, computation of the exact Hessian matrix $H$ is practically infeasible. Previous research (Kwon et al., 2022) shows that the Hessian can be approximated by the empirical Fisher information matrix $\mathcal{I}$ computed as the expectation over the entire dataset by

$$\mathcal{I} := \mathbb{E}\left[\frac{\partial}{\partial W}\mathcal{L}_F(W)\frac{\partial}{\partial W}\mathcal{L}_F(W)^T\right]. \tag{7}$$

In practice, we can also assume $\mathcal{I}$ to be a diagonal matrix. This further simplifies Eq. (6) as

$$\mathcal{L}_F(Q(W)) - \mathcal{L}_F(W) \propto \Delta^T \mathcal{I} \Delta = \sum_i \Delta_i^2 \|\partial \mathcal{L}_F(W)/\partial w_i\|_2^2 = \sum_i \Delta_i^2 \mathcal{I}_{ii}, \tag{8}$$

where the latter result represents a sum of Fisher trace elements $(\mathrm{tr}(\mathcal{I}) = \sum_i \mathcal{I}_{ii})$ weighted by the squared quantization error and it is performed over each element $w_i$ of $W$.

Eq. (8) provides a feasible yet effective sensitivity metric to estimate the impact of quantization noise on the critical objectives. It analytically connects the quantization-caused weight perturbation with the maximum likelihood estimator for critical objectives using Fisher information framework (Ly et al., 2017). Hence, an objective with the larger sensitivity metric leads to inferior critical performance. To verify Eq. (8), we plot the Fisher trace for the critical objectives when estimated using the pretrained floating-point DETR in Fig. 2 (left). We can clearly see a correlation between a larger Fisher trace and a lower critical-category performance metric for the post-training quantization setting.

**Insight 2: Sharp loss landscape leads to a poor generalization for critical categories after quantization-aware training.** During the conventional QAT process, weights of the DETR model are trained to minimize the overall objective $\mathcal{L}_A(Q(W))$. However, a convergence of $\mathcal{L}_A$ does not guarantee good performance on all critical objectives $\mathcal{L}_F$. Foret et al. (2021) find a positive correlation between the validation performance for the objective $\mathcal{L}_F$ and a sharpness $\mathcal{S}$ of the loss landscape around the local minima $Q(W)$ of the QAT. The minima sharpness $\mathcal{S}(Q(W))$ of the quantized model is formulated as

$$\mathcal{S}(Q(W)) = \max_{\|\epsilon\|_2 \leq \rho} \mathcal{L}_F(Q(W) + \epsilon) - \mathcal{L}_F(Q(W)), \tag{9}$$

where $\rho > 0$ is a $\ell_2$ norm bound for the worst-case weight perturbation.

Finding the exact solution to the maximization in Eq. (9) can be computationally costly. With the detailed derivations in Appendix A, we can simplify this problem as

$$\mathcal{S} \approx \max_{||\epsilon||_2 \leq \rho} \left[\mathcal{L}_F(Q(W)) + \epsilon^T g\right] - \mathcal{L}_F(Q(W)) = \max_{||\epsilon||_2 \leq \rho} \epsilon^T \frac{\partial}{\partial W}\mathcal{L}_F(Q(W))$$

$$\propto \frac{\partial}{\partial W}\mathcal{L}_F(Q(W))^T \frac{\partial}{\partial W}\mathcal{L}_F(Q(W)) = \mathrm{tr}(\mathcal{I}). \tag{10}$$

Hence, the trace of the diagonal Fisher information matrix from Eq. (7) approximates the sharpness of the critical loss landscape for the quantized model. The sharper loss landscape leads to inferior test-time critical performance after QAT. To verify this, we empirically measure the Fisher trace using Eq. (10) on different critical objectives of the DETR model after 50 epochs of QAT. As shown in Fig. 2 (right), lower Fisher trace leads to higher critical mAP after QAT. Hence, Fisher trace of the critical objective is a good indication of the model's post-QAT generalization.

### 4.2 FISHER-AWARE MIXED-PRECISION QUANTIZATION SCHEME

With the derived quantization impact on the loss in Eq. (8), we propose a mixed-precision quantization scheme that minimizes the quantization effects within a model-size budget. Specifically, we define the following minimization problem

$$\min_{q_{1:L}} \sum_{i=1}^{L} \Delta_i^2 \left\| \frac{\partial}{\partial w_i} \left( \alpha \mathcal{L}_A(W) + \mathcal{L}_F(W) \right) \right\|_2^2, \text{ s.t. } \sum_{i=1}^{L} q_i \left\| w_i \right\|_0 \leq B, \tag{11}$$

where $L$ denotes the number of layers in the model, $w_i$ is the $i$-th layer's weight vector, and $\Delta_i = Q(w_i) - w_i$ is the quantization error of the $i$-th layer's weight when quantized to $q_i$ bits. $B$ denotes the model size allowance of the quantized model. As the quantization precision $q_i$ takes discrete integer values, the optimization problem in Eq. (11) can be efficiently solved as an Integer Linear Programming (ILP) problem (Dong et al., 2020; Yao et al., 2021).

Note that in Eq. (11) we consider the Fisher information of both critical and overall objectives. This approach allows us to achieve good overall performance and increase the critical performance of interest for the quantized model. A hyperparameter $\alpha$ is used to balance $\mathcal{L}_F$ and $\mathcal{L}_A$.

### 4.3 FISHER TRACE FOR QAT REGULARIZATION

Besides using Fisher information metric to design quantization scheme, we further apply regularization during the QAT process of the quantized DETR model to achieve better generalization performance on critical categories. Following the derivation in Eq. (10), we propose to minimize critical loss sharpness $\mathcal{S}(Q(W))$ during the conventional QAT optimization. Specifically, for a critical objective $\mathcal{L}_F$, we add the Fisher trace regularization as

$$\min_W \mathcal{L}_A(Q(W)) + \lambda \operatorname{tr}(\mathcal{I}_F), \tag{12}$$

where $\lambda \geq 0$ is the strength of the regularization, and $\mathcal{I}_F$ denotes the Fisher information matrix of the critical objective $\mathcal{L}_F(Q(W))$ w.r.t. weight $W$.

In addition to the DETR training loss terms from Eqs. (2) and (3), we further add a distillation loss (Hinton et al., 2015) between the quantized model (student) and the pretrained full-precision model (teacher) following common QAT practice (Dong et al., 2020; Yang et al., 2021). The distillation objective consists of a KL-divergence loss for class logits of the student and teacher models, and a $\ell_1$ loss for the corresponding bounding box coordinates. Since we expect the student model to have the same behavior as the teacher model, the distillation loss uses a fixed one-to-one mapping between the predicted boxes of the two models without performing the Hungarian matching.

## 5 EXPERIMENTS

### 5.1 EXPERIMENTAL SETUP

**Datasets and metrics.** We follow DETR (Carion et al., 2020) setup and use two variants of the COCO 2017 dataset (Lin et al., 2014): COCO detection and COCO panoptic segmentation. The detection dataset contains 118K training images and bounding box labels with 80 categories combined into 12 super categories. The panoptic dataset consists of 133K training examples and corresponding labels with 133 categories and 27 super categories. Both variants contain 5K data points in the validation set. We perform Fisher evaluation and QAT using these training datasets and report the model performance on the validation set. We show both the overall and critical mAP in our experiments. As in Sec. 3.3, we define critical performance on each super category by considering all categories within it as critical and the rest of the categories as non-critical. All mAP reported in the tables are in percentage points. For COCO panoptic we report the mAP of box detection as $\text{mAP}_{\text{box}}$.

**Model architectures.** We conduct the majority of our experiments on the DETR model with ResNet-50 backbone (DETR-R50). To show the scalability, we also experiment with ResNet-101 backbone variant (DETR-R101), DAB-DETR (Liu et al., 2022) and Deformable DETR (Zhu et al., 2021).

**Implementation details.** We perform quantization of the pretrained models using their publicly available checkpoints. We apply symmetric layer-wise weight quantization using Eq. (4), where

Table 3: Comparing critical mAP of different quantization schemes on COCO detection dataset.

| Model | Quant scheme | Critical 4-bit mAP | | | Critical 6-bit mAP | | |
|---|---|---|---|---|---|---|---|
| | | Person | Animal | Indoor | Person | Animal | Indoor |
| DETR-R50 | Uniform | 34.6 | 37.2 | 39.6 | 37.3 | 40.0 | 42.4 |
| | HAWQ-V2 | 35.31±0.05 | 37.90±0.16 | 40.20±0.18 | 37.29±0.04 | 40.20±0.06 | 42.60±0.06 |
| | Fisher-Overall | 35.35±0.04 | 37.96±0.17 | 40.20±0.17 | 37.58±0.11 | 40.74±0.07 | 43.10±0.08 |
| | Fisher-Critical | **35.56**±0.08 | **38.10**±0.09 | **40.33**±0.04 | **37.73**±0.02 | **40.86**±0.06 | **43.26**±0.07 |
| DETR-R101 | Fisher-Overall | 36.36 | **39.30** | 41.70 | 39.1 | 42.0 | 44.4 |
| | Fisher-Critical | **36.42** | 39.23 | **41.80** | **39.2** | **42.5** | **44.9** |
| DAB DETR-R50 | Uniform | 22.32±0.01 | 25.68±0.01 | 27.60±0.01 | 26.24±0.02 | 29.76±0.01 | 31.88±0.01 |
| | HAWQ-V2 | 8.26±0.00 | 11.66±0.00 | 12.80±0.00 | 19.10±0.00 | 19.90±0.00 | 21.60±0.00 |
| | Fisher-Overall | 22.82±0.01 | 27.02±0.00 | **28.96**±0.00 | 26.06±0.01 | 29.20±0.01 | 31.32±0.00 |
| | Fisher-Critical | **23.18**±0.00 | **27.86**±0.22 | 27.98±0.00 | **26.38**±0.00 | **29.28**±0.00 | **31.88**±0.20 |
| Deformable DETR-R50 | Uniform | 28.9 | 32.8 | 34.3 | 46.0 | 49.1 | 51.4 |
| | Fisher-Overall | 42.7 | 46.2 | 48.4 | 46.3 | **49.5** | 51.8 |
| | Fisher-Critical | **43.1** | **46.3** | **48.8** | **46.6** | **49.5** | **52.0** |

Table 4: Comparing critical mAP$_{box}$ of different quantization schemes on COCO panoptic dataset.

| Model | Quant scheme | Critical 4-bit mAP | | | Critical 5-bit mAP | | |
|---|---|---|---|---|---|---|---|
| | | Person | Animal | Indoor | Person | Animal | Indoor |
| Segm. DETR-R50 | Uniform | 8.5 | 11.4 | 12.4 | 8.9 | 13.7 | 16.0 |
| | Fisher-Overall | 16.64 | 21.60 | 23.80 | 18.79 | 24.00 | 26.70 |
| | Fisher-Critical | **16.68** | **21.69** | **23.85** | **19.05** | **24.15** | **26.87** |

weights are scaled by the max of absolute values without clamping. We keep normalization and softmax operations at full precision. For sensitivity analysis, we compute Fisher trace for our method using all training set. Whereas for the implementation of HAWQ-V2 (Dong et al., 2020) baseline, we randomly sample 1,000 training images due to the high cost of Hessian computation. We solve mixed-precision quantization problem in Eq. (11) by the ILP with 3-8 bit budget for each layer. We perform QAT with the straight-through gradient estimator (Bengio et al., 2013) for 50 epochs with 1e-5 learning rate. Regularization strength $\lambda$ in Eq. (12) grows linearly from 1e-3 to 5e-3 throughout the training epochs when Fisher regularization is applied. To mitigate the variance in training, in all experiments we report the mean and, if shown, $\pm$ standard error of the final 5 epochs of training.

## 5.2 PERFORMANCE OF FISHER-AWARE QUANTIZATION SCHEME

As proposed in Sec. 4.2, we use Fisher information as a sensitivity measurement for designing mixed-precision quantization scheme for object detection. We compare the proposed quantization scheme with linear uniform quantization (Polino et al., 2018) and HAWQ-V2 (Dong et al., 2020) on the COCO dataset task using its detection and panoptic variants. Additional CityScapes (Cordts et al., 2016) dataset results are reported in Appendix C. Tab. 3 reports the critical mAP of super category "Person", "Animal", and "Indoor" for the COCO detection dataset. We generate HAWQ-V2 and Fisher-Overall quantization schemes with only the overall objective, and evaluate the model after QAT on these super categories. For Fisher-Critical schemes, we apply the critical-category objective for the super category of interest to the ILP optimization in Eq. (11), and report the critical mAP of the quantization scheme corresponding to each super category. The overall mAP is not affected much by the choice of Fisher-Critical vs. Fisher-Overall objectives as additionally described in Appendix C.

With the same average quantization precision, our Fisher-aware method consistently outperforms uniform quantization and the mixed-precision scheme derived from HAWQ-V2 on different models and datasets. We note that the improvement of HAWQ-V2 over the uniform quantization is not consistent on DETR-based models. This is caused by the instability of Hessian trace estimation for the complicated DETR architecture and the harder object detection task. Fisher-aware approach, on the other hand, is stable. In addition, we evaluate the time to estimate the Fisher trace w.r.t. the Hessian trace for a batch of images on a Tesla P100 GPU, where Fisher trace can be estimated with 200-300× less computational cost. This allows us to estimate Fisher trace with a larger amount of training data, which leads to a higher precision and stability.

Comparing to Fisher-overall, applying the critical objective leads to consistent improvements in the corresponding critical performance. In particular, we improve critical mAP by up to 0.2% on

Table 5: Comparing performance of DETR-R50 model on COCO detection dataset with different QAT objectives. Fisher-critical quantization scheme with Person category is used for all models.

| Model | QAT objective | 4-bit mAP | | 6-bit mAP | |
|---|---|---|---|---|---|
| | | Overall | Person | Overall | Person |
| DETR-R50 | Overall | **37.07**±0.07 | 35.56±0.08 | 39.67±0.10 | 37.73±0.02 |
| | Fisher Reg | 36.97±0.06 | **35.75**±0.04 | **39.70**±0.08 | **37.78**±0.01 |

Table 6: Comparing $mAP_{box}$ for DETR-R50 model on COCO panoptic dataset with different QAT objectives. Fisher-critical quantization scheme with Person category is used for all models.

| Model | QAT objective | 4-bit mAP | | 5-bit mAP | |
|---|---|---|---|---|---|
| | | Overall | Person | Overall | Person |
| Segm. | Overall | 33.24±0.10 | 16.68±0.01 | 36.08±0.07 | 19.05±0.09 |
| DETR-R50 | Fisher-Reg | **33.29**±0.05 | **16.79**±0.03 | **36.12**±0.06 | **19.39**±0.11 |

DETR-R50, 0.5% on DETR-R101, 0.8% on DAB DETR-R50, and 0.4% on Deformable DETR-R50. Note that the originally poorly-performing critical categories such as "Person" result in significant improvements. We further extend our experiments to the COCO panoptic dataset with the same setup, where we report the $mAP_{box}$ of different quantization schemes in Tab. 4. With the increased number of categories in the panoptic dataset, the impact of quantization on each individual critical category becomes even more disparate. Fisher-aware quantization with the overall objective improves critical mAP by about $2\times$ over uniform quantization. Further improvement on critical mAP is consistently achieved with the Fisher-critical quantization scheme. This further shows the importance of considering critical objectives when applying object detection models to real-world applications.

### 5.3 PERFORMANCE OF FISHER-TRACE REGULARIZATION

In this section, we compare the post-QAT results when using the conventional overall loss $\mathcal{L}_A$ only vs. our approach with Fisher trace regularization from Sec. 4.3 on the COCO detection dataset in Tab. 5. The experimental results show that applying the proposed Fisher-trace regularization further improves critical mAP. When combined with the mixed-precision quantization scheme from Sec. 4.2, our method leads to a 1.15% and 0.48% critical ("person" class) performance improvement on DETR-R50 model over the uniform quantization in Tab. 3 for 4-bit ($34.6\% \rightarrow 35.75\%$) and 6-bit ($37.3\% \rightarrow 37.78\%$) precision, respectively. Note that our regularization scheme has a negligible impact on the overall mAP: $37.07\% \rightarrow 36.97\%$ for 4-bit and $39.67\% \rightarrow 39.70\%$ for 6-bit precision, respectively.

Tab. 6 reports post-QAT results for COCO panoptic dataset. The proposed regularization further increases critical performance by 0.11% and 0.34% mAP for, correspondingly, 4-bit and 5-bit precision settings when compared to our PTQ results in Tab. 4. Note that the uniform PTQ quantization significantly underperforms in this setting. In Appendix D ablation study, we analyze the impact of regularization strength. In addition, we compare model generalization abilities for a naive approach when the critical loss $\mathcal{L}_F$ is added to the overall loss $\mathcal{L}_A$ as a heuristic QAT objective, and our Fisher-trace regularization scheme that minimizes sharpness of the loss landscape.

## 6 CONCLUSIONS

This work investigated the impact of quantization on the fine-grained critical-category performance of DETR-based object detectors. Motivated by the demand for practical applications, we formulated the critical performance via the logit-label transformation of the corresponding categories. We found that both the conventional PTQ and QAT cause disparate quantization effects on such critical performance. We theoretically linked the disparate quantization effects with the sensitivity of critical objectives to the quantization weight perturbation and the sharpness of the critical loss landscape in the QAT. We characterized both derivations with the trace of the Fisher information matrix of the critical objectives w.r.t. model weights. We proposed the Fisher-aware mixed-precision quantization scheme and Fisher-trace regularization to improve the critical performance of interest. We hope this work motivates future explorations on the fine-grained impacts of other compression methods in the computer vision area and a general machine learning research.

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

In the following supplementary materials, we provide detailed derivation and additional visualizations and experimental results. Specifically, Appendix A provides detailed derivation on Eq. (10). Appendix B shows detailed visualizations of the layer-wise Fisher sensitivity and the corresponding quantization precision settings derived using the proposed Fisher-aware quantization scheme method. Appendix C further analyzes how the overall and critical-category objectives from Eq. (11) impact the assigned precision, and shows additional experimental results for the Fisher-aware quantization scheme. For Fisher-trace regularization, Appendix D performs ablation study on the impact of regularization strength $\lambda$, compares the critical Fisher trace of the converged model with or without Fisher-trace regularization, and verifies the design choice of using Fisher regularization instead of the sum of overall and critical-category objective in the optimization.

## A  DETAILED DERIVATION OF EQ. (10)

Starting from the formulation in Eq. (9), we perform the first-order Taylor expansion on the perturbed loss $\mathcal{L}_F(Q(W) + \epsilon)$ as

$$\mathcal{L}_F(Q(W) + \epsilon) \approx \mathcal{L}_F(Q(W)) + \epsilon^T \frac{\partial}{\partial W} \mathcal{L}_F(Q(W)). \tag{13}$$

By substituting Eq. (13) into Eq. (9), the solution to maximization problem can be simplified as

$$\mathcal{S}(Q(W)) = \max_{||\epsilon||_2 \leq \rho} \mathcal{L}_F(Q(W) + \epsilon) - \mathcal{L}_F(Q(W)) \approx \max_{||\epsilon||_2 \leq \rho} \epsilon^T \frac{\partial}{\partial W} \mathcal{L}_F(Q(W)). \tag{14}$$

Note that both $\epsilon$ and $\partial \mathcal{L}_F(Q(W))/\partial W$ are vectors with the same dimensions as weight $W$. Then, their inner product can achieve the maximum when they are parallel vectors. Therefore, we can solve the maximization in Eq. (14) as

$$\begin{aligned}
\mathcal{S} &\approx \max_{||\epsilon||_2 \leq \rho} \epsilon^T \frac{\partial}{\partial W} \mathcal{L}_F(Q(W)) \\
&= \frac{\rho}{\|\frac{\partial}{\partial W} \mathcal{L}_F(Q(W))\|_2} \frac{\partial}{\partial W} \mathcal{L}_F(Q(W))^T \frac{\partial}{\partial W} \mathcal{L}_F(Q(W)) \\
&\propto \frac{\partial}{\partial W} \mathcal{L}_F(Q(W))^T \frac{\partial}{\partial W} \mathcal{L}_F(Q(W)) = \mathrm{tr}(\mathcal{I}),
\end{aligned} \tag{15}$$

which is the final approximation of the loss landscape sharpness in Eq. (10).

## B  DETAILS ON FISHER-AWARE QUANTIZATION SCHEMES

This section illustrates the Fisher-aware sensitivity and the corresponding quantization schemes assigned to the Fisher-overall configurations from Tab. 3 and Tab. 4. We report four models, namely DETR-R50 (Fig. 3), DETR-R101 (Fig. 4), DAB DETR-R50 (Fig. 5), and Deformable DETR-R50 (Fig. 6), on the COCO detection dataset, and, additionally, report DETR-R50 (Fig. 7) and DETR-R101 (Fig. 8) on the COCO panoptic dataset.

As shown in the figures, the backbone layers demonstrate a relatively stable sensitivity distribution, while the transformer encoder and decoder layers show sensitivity distribution with high variance. This is expected given the different functionalities of transformer layers within an attention block (Carion et al., 2020). Correspondingly, the quantization precision is assigned to each layer via solving the ILP in Eq. (11), where higher precision is assigned to layers with higher sensitivity, while satisfying the constraint on the overall budget allowance.

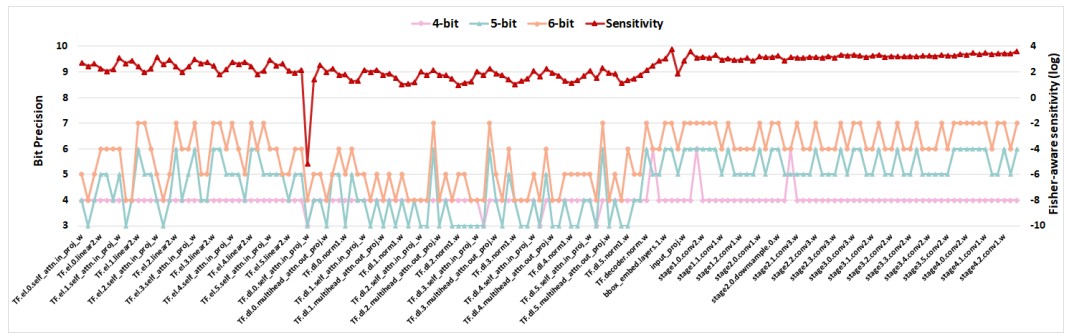

Figure 3: Bit precision vs. layer-wise sensitivity for DETR-R50 on COCO detection dataset.

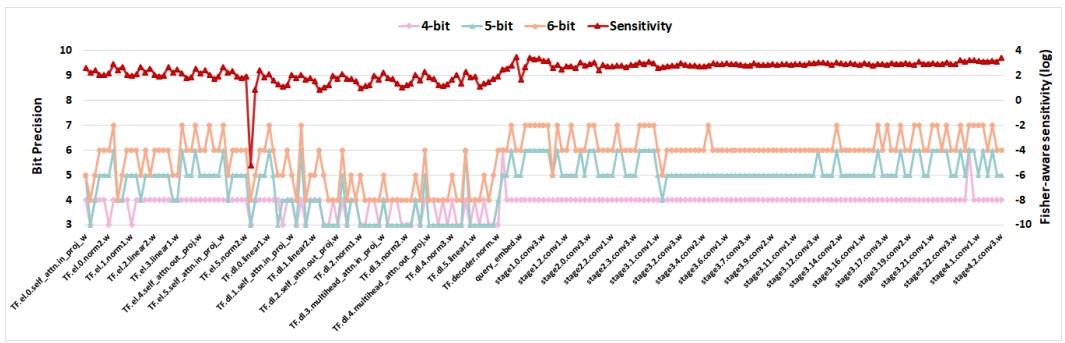

Figure 4: Bit precision vs. layer-wise sensitivity for DETR-R101 on COCO detection dataset.

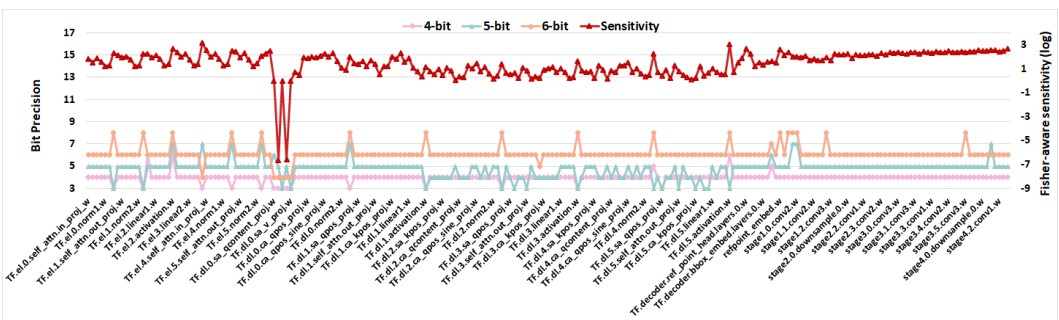

Figure 5: Bit precision vs. layer-wise sensitivity for DAB DETR-R50 on COCO detection dataset.

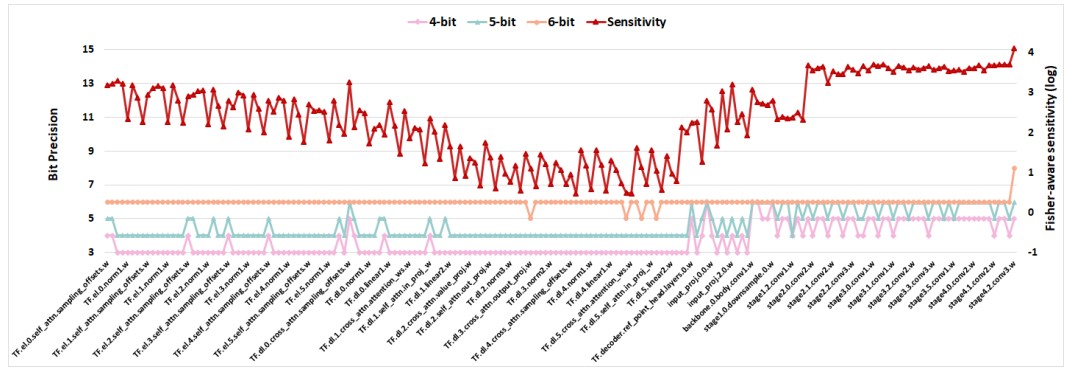

Figure 6: Bit precision vs. layer-wise sensitivity for Deformable DETR-R50 on COCO detection dataset.

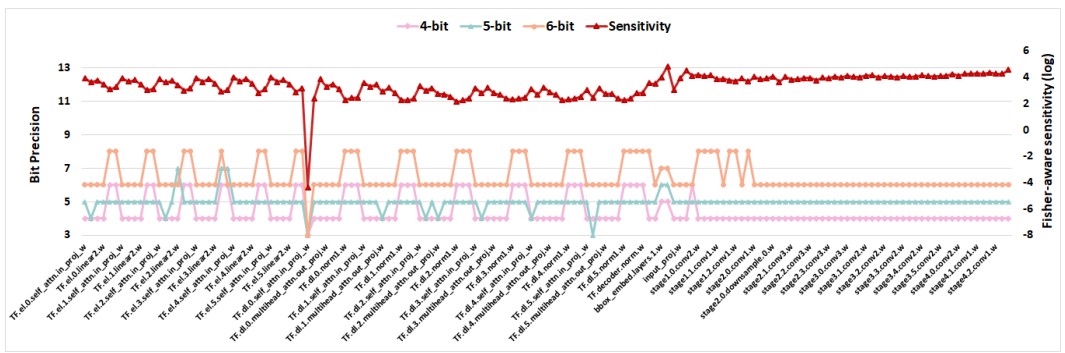

Figure 7: Bit precision vs. layer-wise sensitivity for DETR-R50 on COCO panoptic dataset.

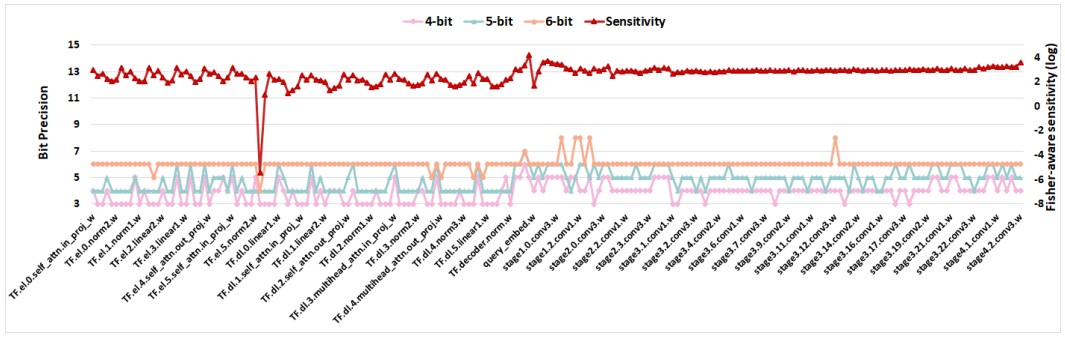

Figure 8: Bit precision vs. layer-wise sensitivity for DETR-R101 on COCO panoptic dataset.

## C    ADDITIONAL RESULTS ON FISHER-AWARE QUANTIZATION SCHEME

This section provides additional experimental results on utilizing critical-category objectives in the proposed Fisher-aware mixed-precision quantization scheme. To start with, Figs. 9 to 14 compares the quantization scheme of Fisher-overall configuration and Fisher-critical configuration for different models and critical categories reported in Tabs. 3 and 4. As shown in the figures, the inclusion of different critical objectives into the sensitivity analysis leads to significant change in the precision assigned to some of the layers. This further illustrates the diverse sensitivity distribution of various critical-category objectives across layers. By considering the proposed objectives of interest in the sensitivity analysis, we are able to improve the critical performance of quantized models.

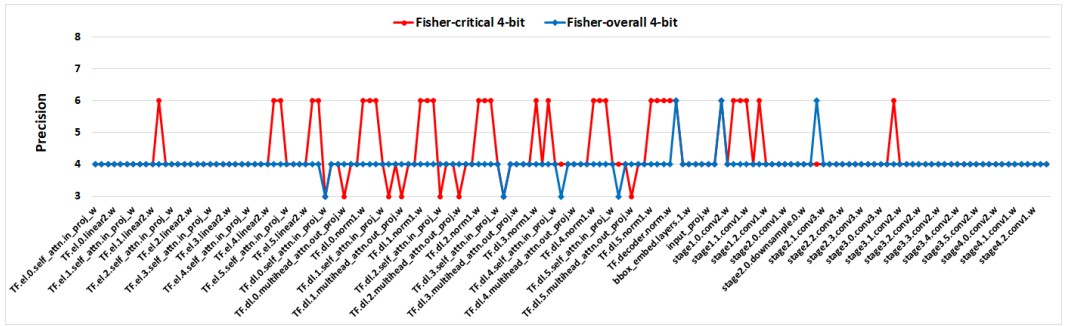

Figure 9: Comparison of Fisher-critical and Fisher-overall mixed-precision schemes for DETR-R50 on COCO detection dataset when applied to **person category**.

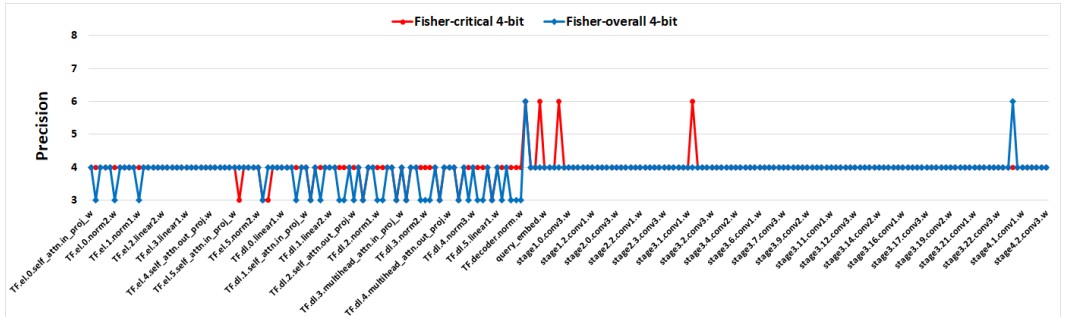

Figure 10: Comparison of Fisher-critical and Fisher-overall mixed-precision schemes for DETR-R101 on COCO detection dataset when applied to **indoor category**.

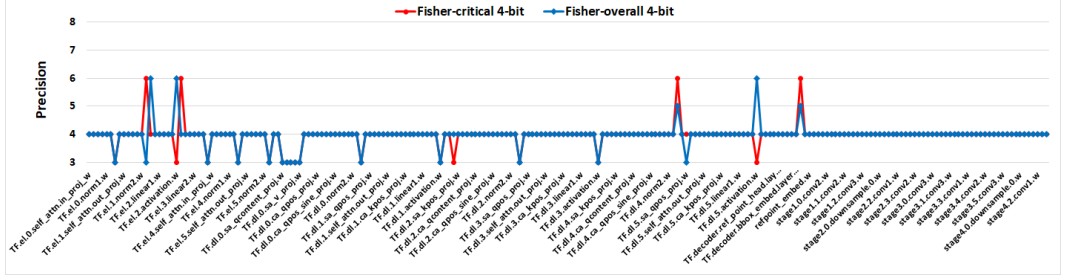

Figure 11: Comparison of Fisher-critical and Fisher-overall mixed-precision schemes for DAB DETR-R50 on COCO detection dataset when applied to **animal category**.

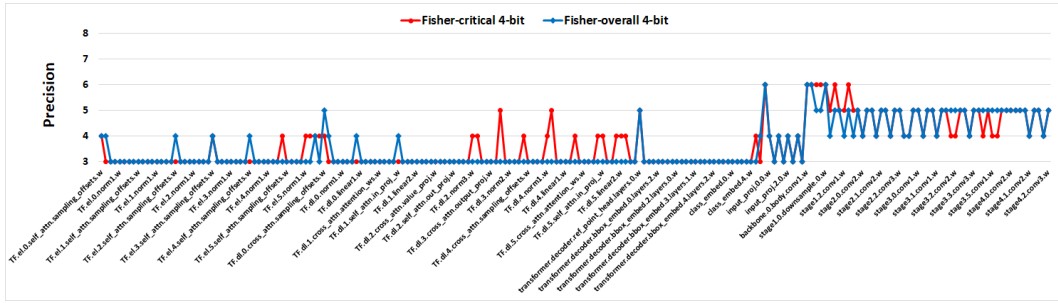

Figure 12: Comparison of Fisher-critical and Fisher-overall mixed-precision schemes for Deformable DETR-R50 on COCO detection dataset when applied to **person category**.

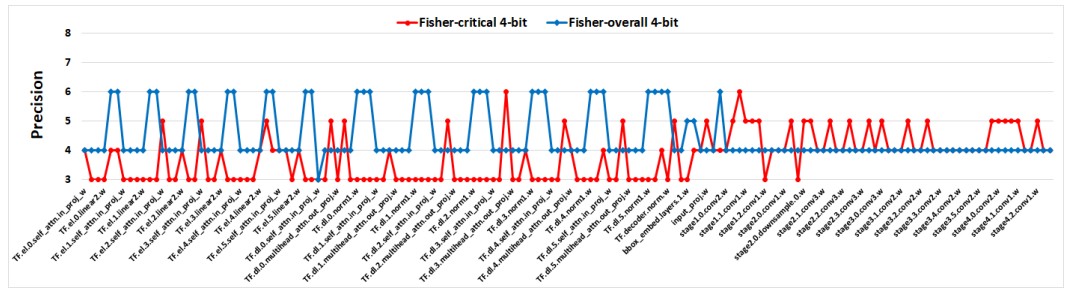

Figure 13: Comparison of Fisher-critical and Fisher-overall mixed-precision schemes for DETR-R50 on COCO panoptic dataset when applied to **animal category**.

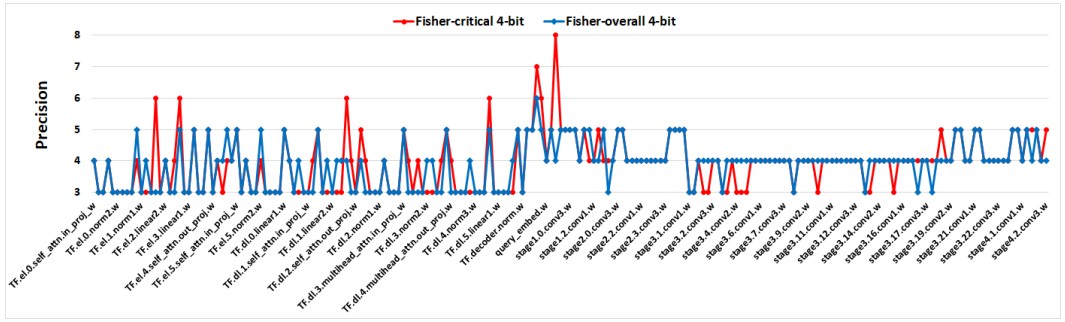

Figure 14: Comparison of Fisher-critical and Fisher-overall mixed-precision schemes for DETR-R101 on COCO panoptic dataset when applied to **person category**.

As Tab. 3 contains only the critical-category metrics, here we report the overall mAP of these quantization schemes in Tabs. 7 and 8. They show the impact of the proposed objectives on the overall performance. In general, Fisher-critical quantization scheme leads to a similar overall performance with the Fisher-overall scheme, and they both are significantly better than the conventional uniform and HAWQ-V2 quantization schemes. In some cases, the improvement of critical performance with the Fisher-critical quantization scheme also improves the overall performance. This indicates that the addition of such critical-category objective in the sensitivity analysis can be useful for increasing the overall performance as well. This is an interesting direction to explore in future work.

Table 7: Overall mAP on COCO detection dataset for 4-bit setting.

| Model | Uniform | Fisher-overall | Fisher-Person | Fisher-Animal | Fisher-Indoor |
|---|---|---|---|---|---|
| DETR-R50 | 36.7 | **37.12**±0.06 | 37.07±0.07 | 37.01±0.11 | 36.99±0.05 |
| DETR-R101 | 37.4 | **38.26** | 38.22 | 37.97 | 38.24 |
| DAB DETR-R50 | 22.7 | 24.42±0.00 | 25.28±0.00 | **25.84**±0.27 | 24.08±0.00 |
| Deformable DETR-R50 | 28.8 | 44.1 | **44.5** | 44.1 | **44.5** |

Table 8: Overall mAP on COCO detection dataset for 6-bit setting.

| Model | Uniform | Fisher-overall | Fisher-Person | Fisher-Animal | Fisher-Indoor |
|---|---|---|---|---|---|
| DETR-R50 | 39.4 | 39.57±0.10 | **39.67**±0.10 | 39.60±0.04 | 39.61±0.08 |
| DETR-R101 | 39.2 | 41.8 | 41.8 | **42.1** | **42.1** |
| DAB DETR-R50 | 28.00 | 27.20±0.00 | **28.42**±0.00 | 27.30±0.00 | 27.94±0.15 |
| Deformable DETR-R50 | 47.8 | 48.1 | **48.5** | 48.1 | **48.5** |

Finally, we provide additional results of the Fisher-aware quantization scheme with the CityScapes dataset in Tab. 9. We perform object detection task with DETR model on the CityScapes dataset following the settings of Wang et al. (2022)[3]. We can observe the same trend that the proposed Fisher-Overall scheme significantly outperforms uniform quantization, whereas Fisher-Critical scheme further improves the performance on corresponding critical categories.

Table 9: Critical-category mAP on CityScapes for DETR-R50 with different quantization schemes.

| Precision | Quant. scheme | All mAP | Critical mAP | | | |
|---|---|---|---|---|---|---|
| | | | Construct | Object | Human | Vehicle |
| FP | - | 11.7 | 8.7 | 17.8 | 18.0 | 19.0 |
| | Uniform | 5.2 | 3.6 | 8.7 | 8.8 | 9.2 |
| 4-bit | Fisher-Overall | 8.8 | 5.5 | 12.6 | 13.8 | 14.6 |
| | Fisher-Critical | **9.0** | **6.5** | **13.7** | **14.0** | **14.7** |

# D  ABLATION STUDY ON FISHER-TRACE REGULARIZATION

We start with the discussion about the impact of regularization strength $\lambda$ on the overall and the critical performance in the QAT process. Similarly to previous work on regularized training (Yang et al., 2022), $\lambda$ controls the tradeoff between the overall performance and the generalization gap for the critical objective. Tab. 10 shows the overall and critical mAP during training if we set $\lambda$ to a smaller value, i.e. 1e-3. It can be seen that the Fisher trace regularization significantly improves critical mAP during epoch range from 20 to 30 (up to 0.5%). However, as the training progresses towards convergence, the critical performance drops while overall performance increases, indicating the occurrence of overfitting. However, setting $\lambda$ too large (i.e. 5e-3) in the initial epochs of the QAT process significantly affects the convergence of the overall training objective. These observations indicate that during the QAT process, a smaller regularization is needed initially to facilitate better convergence, while a larger regularization is needed towards the end to prevent overfitting. To this end, in this work we utilize a linear scheduling of the regularization strength as discussed in Sec. 5.1, which can be formulated as $\lambda = \max\left[\lambda_T \frac{t}{T}, \lambda_0\right]$, where $t$ is the current epoch, $T$ is the total number of epochs, and $\lambda_0, \lambda_T$ denote the initial and final regularization strength, respectively. This scheme leads to better critical mAP after the convergence as shown in Tabs. 4 and 5.

---

[3] https://github.com/encounter1997/DE-DETRs

Table 10: QAT performance of DETR-R50 on COCO detection dataset. Person category is considered as critical, and 4-bit Fisher-Critical quantization scheme is applied. The mAP metrics at each epoch are reported using overall/critical format.

| $\lambda$ | Epoch 10 | Epoch 20 | Epoch 30 | Epoch 40 | Epoch 50 |
|---|---|---|---|---|---|
| 0 | 36.8/34.8 | 36.7/34.7 | 36.9/34.9 | 37.3/35.1 | 37.2/35.2 |
| 1e-3 | 36.5/34.5 | 36.9/**35.2** | 36.8/**35.3** | 37.1/35.0 | 37.2/35.1 |

Table 11: Fisher trace of the critical objective when applied to DETR-R50 on COCO detection dataset. In this setting, the person category is considered as critical.

| Precision | Quant. scheme | Reg. | Fisher trace |
|---|---|---|---|
| | Uniform | No | 37.3K |
| 4-bit | Fisher-critical | No | 30.4K |
| | Fisher-critical | Yes | **14.9K** |
| | Uniform | No | 88.9K |
| 6-bit | Fisher-critical | No | 18.2K |
| | Fisher-critical | Yes | **15.5K** |

To further show the effectiveness of the Fisher-trace regularization, we compute the Fisher trace of the critical objective on the quantized DETR model after QAT. We compare the critical Fisher trace of models with different quantization and training schemes in Tab. 11 with 10,000 data examples sampled from the COCO training set. Both the 4-bit and 6-bit uniform quantization settings lead to the largest Fisher trace on the critical objective, while our Fisher-aware mixed-precision quantization scheme helps to reduce the trace after QAT. Furthermore, the proposed regularization scheme with explicit control over the Fisher trace results in the lowest value that indicates the least sharp local minima. This observation confirms our insight in Sec. 4.1, where large critical Fisher trace leads to inferior critical performance.

Finally, we verify the necessity of applying Fisher-trace regularization for the QAT of DETR model. Specifically, we compare our proposed scheme with performing regular QAT with the summation of overall and critical objective. For the Fisher-trace regularization, the motivation comes from our Insight 2 in Sec. 4.1, where the QAT gap is caused by the **sharp loss landscape leading to a poor generalization**. This cannot be simply resolved with the addition of critical objectives in the training loss, as illustrated by the results in Tab. 12.

Table 12: QAT performance of DETR-R50 on COCO detection (left) and COCO panoptic (right) datasets. All models are quantized with the corresponding Fisher-critical scheme. 4-bit is used for COCO detection and 5-bit for COCO panoptic dataset, respectively.

| QAT objective | All mAP | Person mAP | QAT objective | All mAP | Person mAP |
|---|---|---|---|---|---|
| Overall | 37.07±0.07 | 35.56±0.08 | Overall | 36.08±0.07 | 19.05±0.09 |
| Overall+Critical | 37.11±0.04 | 35.39±0.06 | Overall+Critical | 36.07±0.05 | 19.19±0.10 |
| Fisher reg | 36.97±0.06 | **35.75±0.04** | Fisher reg | **36.12±0.06** | **19.39±0.11** |

