# OpenReview forum: "Fisher-aware Quantization for DETR Detectors with Critical-category Objectives"
_ICLR.cc/2024/Conference — Submitted to ICLR 2024_

### Official Review · Reviewer_DjvC · 2023-10-17

**Soundness:** 2 fair
**Presentation:** 3 good
**Contribution:** 2 fair
**Rating:** 6
**Confidence:** 3

**Summary:**

In this paper, the authors first argure that rather than the overall performance, the fine-grained critical-category performance should be more focused during quantization. They then link the fine-grained critical-category performance with the Fisher information and propose to perform quantization in a Fisher-information aware manner.

**Strengths:**

1. The problem that this paper points out seems to be reasonable and interesting.

2. The paper is easy to follow.

**Weaknesses:**

(See questions below for details)

**Questions:**

1. My major concern w.r.t. this paper is that, while I agree on the importance of the proposed problem that we should particularly focus on certain classes instead of all classes, I feel that this can be achieved in many ways. For example, during training, one can particularly control the loss weight for certain particular classes. Moreover, during quantization, one can perform input-aware bit assignment [1] to set higher bit for images in which critical classes are hardly distinguishable and lower bit for other images. Thus, while I admit that the Fisher information can be a feasiable perspective, I hope that the authors can discuss more on its advantages over other seems feasible ways.

[1] Instance-Aware Dynamic Neural Network Quantization

2. I am worried on the generalizability of the proposed method. On the one hand, in real-life scenarios, there are often cases that the user only know that there is an object detector but does not know that whether the object detector is based on DETR or based on other object detectors. Thus, if this method can only be used on DETR, this can limit its generalizability and usage. On the other hand, the observation and demonstration in this paper seem to be focus on COCO, which can also be limited especially when this paper emphasize on its solving of real-world problem. I guess a real-world problem can only be regarded as well-solved if it can be solved in a rather general manner.

3. The last concern I have is w.r.t. the experiment. Specifially, from my perspective, there can exist a mismatch between the experiment and the claim of this paper. In other words, the improvement of Fisher-Critical over Fisher-Overall or even Uniform is often limited. This made me have the feeling that, this paper fails to convince me that it proposes a good method for solving the "the fine-grained critical-category performance" problem.

---

> ### Author Response · Authors · 2023-11-20
> **Response to Reviewer Djvc**
>
> We thank reviewer Djvc for your thorough review and valuable comments. Hopefully, the following responses can address your concerns.
>
> &nbsp;
>
> **Q1: Importance of our proposed method**
>
> A1: Thank you for bringing up these possible methods. As we do not claim Fisher-aware quantization to be the only way of improving the fine-grained performance of the quantized model, here we discuss the advantage of our method over other alternatives.
>
> For adjusting the weight of fine-grained losses, we have compared our proposed scheme with performing regular QAT with the summation of overall and critical objectives in Tab. 12 of Appendix D. For the Fisher-trace regularization, the motivation comes from our Insight 2 in Sec. 4.1, where the QAT gap is caused by the sharp loss landscape leading to a poor generalization. This cannot be simply resolved with the addition of critical objectives in the training loss, as illustrated by the results in Tab. 12, where the summed loss does not consistently lead to critical performance improvements.
>
> As for the Instance-aware dynamic quantization method, it may not bring real efficiency when deployed on a hardware system. All the weights need to be stored at the highest possible precision, and the switching of precision at inference time will bring additional overhead. Furthermore, multiple objects with different classes are likely to exist in the same scene for object detection tasks, which brings further challenges for instance-aware methods on deciding the quantization scheme to use. Our proposed method works with a static quantization scheme for all the inputs, which does not suffer from the aforementioned drawbacks.
>
> &nbsp;
>
> **Q2: Generalizability of the proposed method**
>
> A2: This paper focuses on the quantization performance of the DETR model and its variants given their wide usage as state-of-the-art in object detection benchmarks. We also find the quantization of DETR is more challenging and interesting compared to CNN-based models, as the CNN quantization has been extensively studied in the past. Our focus on DETR quantization is also echoed by the recent work Q-DETR [1] in CVPR2023, which validates our motivation.
>
> Within the scope of DETR-based models, we have conducted our experiments with multiple architectures including DETR-R50, DETR-R101, DAB DETR, and Deformable DETR, and on multiple datasets including COCO (Tab. 3 and 5), COCO-Panoptic (Tab. 4 and 6), and CityScapes (Tab. 9). Our proposed method shows consistent improvements across all these settings. Per the suggestion of reviewer ka68, we further evaluate our Fisher-aware quantization scheme with the Q-DETR training objective, and observe similar improvements in the performance. These results verify that our approach is generalizable across architectures, datasets, and QAT methods.
>
> [1] Xu, S.,et al. (2023). Q-DETR: An Efficient Low-Bit Quantized Detection Transformer. In Proceedings of CVPR (pp. 3842-3851).
>
> &nbsp;
>
> **Q3: Significance of the proposed method**
>
> A3: We would like to note that both the Fisher-aware quantization scheme (both overall and critical) and the Fisher-trace regularization on the DETR model are our contributions derived from our theoretical analysis in Sec. 3. When combined, they lead to an improved critical mAP performance over uniform quantization baselines by up to 1.15% on 4-bit DETR-R50 on the COCO dataset, and up to **10.4%** on COCO-Panoptic where the critical performance is hindered more due to the large number of classes (Sec. 5.3).
>
> Also, we would like to argue that the significance of the empirical improvements should not be solely determined by the value itself. Our results of the Fisher-critical quantization scheme are significant in the sense that 1) the improvements are mostly 3x larger than the standard error induced by the training randomness, and 2) the improvement is consistent across different precisions, architectures, and datasets. This verifies the improvements brought by our proposed method are significant and not random. Furthermore, we find the improvement of our method is higher when applied to more complicated architectures. Critical mAP improves by up to 0.5% on DETR-R101, 0.8% on DAB DETR-R50, and 0.4% on Deformable DETR-R50 with Fisher-critical over Fisher-overall, compared to the 0.2% improvement on DETR-R50. This further shows the potential of the proposed method as larger models may require more compression.
>
> Besides the observable improvements brought by our method, we would like to emphasize that this paper is the first to investigate the impact of quantization on the fine-grained critical-category performance of DETR-based object detectors, which is a practical setting yet largely overlooked by the community. As we take an initial step to tackle this challenge in this work, it will motivate future explorations on the fine-grained impacts of other compression methods in the computer vision area and general machine learning research.

---

> > ### Comment · Reviewer_DjvC · 2023-11-21
> >
> > After reading the comments from the author, while my concerns are partially resolved, I am still confused especially w.r.t. Q3. Especially, I find that Reviewer WkhP also seems to have the same concern about the performance.
> >
> > Specifically, in the answers to Q3, the authors seem to claim that they can achieve very significant improvements on COCO-Panoptic. However, this means that, when the proposed method is added to a commonly used detector, the performance improvement is less significant on the detection task, while more significant on the segmentation task. This leads to difficulty in my judgment of the significance of this paper, as I believe the key difference between the two tasks/datasets is much beyond the "large number of classes".
> >
> > Moreover, in the answers to Q3, the authors seem to also emphasize their novelty on "first exploration". However, this claim seems to even deepen my worry on the performance. This is because, we often seem to expect less and harder improvement when a task is near saturation, while when a task is "first explored", we expect improvement of performance to be kind of easier.
> >
> > Overall, I will keep my rate for now. At the same time, I am curious about other reviewers' opinions on the performance.

---

> ### Author Response · Authors · 2023-11-21
> **Clarification on your misunderstanding of COCO-Panoptic**
>
> Thank you for your feedback!
>
> We would like to point out a misunderstanding you had on our reported performance of the COCO-Panoptic dataset. As mentioned in Sec. 5.1, we perform **box detection task** with the COCO-Panoptic dataset and report the mAP box in the results. Here we follow the box detection training process described in the official DETR repo (https://github.com/facebookresearch/detr#training-1).
>
> COCO-Panoptic dataset extends from the 80 categories of COCO to a total of 133 categories grouped into 27 super categories. This significant increase in category numbers leads to a more servere impact on the critical performance. Our methods lead to significant imrovements in this challenging setting. More significant inprovements are also observed on complicated DETR-based architectures, where we see up to 2% critical mAP performance on 4-bit DAB-DETR, and up to **14.5%** on Deformable DETR comparing to Uniform quantization baseline. All these verify the potential of the proposed method.

---

> > ### Comment · Reviewer_DjvC · 2023-11-21
> >
> > Thanks for the author's reply.
> >
> > I think I am convinced this time, especially on the part that more significant improvements are achieved on more complicated architectures. This leads the proposed method to have a particular advantage at least in certain circumstances and I have increased my rate from 5 to 6.

---

> > > ### Author Response · Authors · 2023-11-21
> > >
> > > Thank you very much for your timely feedback and support!

---

### Official Review · Reviewer_ka68 · 2023-10-22

**Soundness:** 3 good
**Presentation:** 3 good
**Contribution:** 3 good
**Rating:** 6
**Confidence:** 3

**Summary:**

• The paper formulates the critical-category performance for object detection applications and observe
disparate effects of quantization on the performance of task-critical categories.

• The paper provides analytical explanations of the quantization effects on critical-category performance for DETR-based models using a theoretical link to the Fisher information matrix.

• The paper proposes a Fisher-aware mixed-precision quantization scheme that considers the sensitivity
of critical-category objectives and improves corresponding detection metrics.

• The paper proposes Fisher-trace regularization for the loss landscape of our objectives during
quantization-aware training to further improve critical-category results.

**Strengths:**

1. The writing of the article meets academic standards;

2. Clear motivation;

3. Strong mathematical analysis

**Weaknesses:**

1. Lack of references to relevant studies

2. There are too few comparisons with other papers

**Questions:**

1. Are you the first to conduct this research? As far as I know Q-DETR （CVPR2023) is the first paper about QUANTIZATION FOR DETR. However, your research contents are different. Q-DETR focuses on how to enhance the performance of quantized DETRs. You focus on  CRITICAL-CATEGORY OBJECTIVES. I think you should cite Q-DETR.

2. I think you should compare it to Q-DETR. Q-DETR's model has been released on Git Hub. Although you are different, I think you are comparable.

---

> ### Author Response · Authors · 2023-11-20
> **Response to Reviewer ka68**
>
> We thank reviewer ka68 for your thorough review and valuable comments. Hopefully, the following responses can address your concerns.
>
> Thank you for bringing up the Q-DETR paper to our attention. We acknowledge the contribution of Q-DETR as a pioneer work in designing better QAT objectives and algorithms for DETR. We are happy to cite the paper in our related work section.
>
> As you have clearly identified, Q-DETR and our paper work on different aspects of DETR quantization. Q-DETR focuses on better distillation objectives to improve the overall DETR performance in QAT. In contrast, our work proposes a Fisher-aware mixed-precision quantization scheme design and a Fisher trace regularization, with a special focus on improving critical-category objectives. In this sense, the contributions of both papers are orthogonal to each other. Following your suggestion, we have included an experiment of training DETR models quantized with different mixed-precision quantization schemes using the Q-DETR distillation objective. Due to the limitation in time, we ran all experiments with the DETR-R50 model on the COCO dataset, with **a total of 50 epochs using a 1e-5 learning rate**. All the other settings are the same as described in Sec. 5.1. Though the reported performance here is lower than that of the Q-DETR paper due to the 10x less training epochs, we believe the trend in relative performance remains if longer training is conducted.
>
> |   Model  |   Quant scheme  | Overall 4-bit mAP | Person 4-bit mAP | Animal 4-bit mAP |
> |:--------:|:---------------:|-------------------|:----------------:|:----------------:|
> | DETR-R50 |     Uniform     |  34.85 $\pm$ 0.04 | 33.00 $\pm$ 0.04 | 35.53 $\pm$ 0.03 |
> |          |  Fisher-Overall |  **35.10** $\pm$ 0.12 | 33.47 $\pm$ 0.09 | 36.08 $\pm$ 0.06 |
> |          | Fisher-Critical |  35.04 $\pm$ 0.11 | **33.72** $\pm$ 0.07 | **36.43** $\pm$ 0.04 |
>
> The results are consistent with our findings in Tab. 3 that the proposed Fisher-overall quantization scheme improves over uniform quantization on the overall model performance, whereas the Fisher-critical quantization scheme further improves the corresponding critical performance. This result verifies that the disparate impact on critical performance exists for different QAT objectives, while our method is generalizable to other QAT objectives.

---

> ### Comment · Reviewer_ka68 · 2023-11-23
>
> I'm shocked that you didn't know about the first and unique article in your field, and it was published in CVPR.

---

> > ### Comment · Reviewer_ka68 · 2023-12-04
> >
> > Although the author did not conduct sufficient research and the performance was not state-of-the-art, this work is the second in the field and is significantly different from the first, so I believe this score is appropriate.

---

### Official Review · Reviewer_WkhP · 2023-11-07

**Soundness:** 3 good
**Presentation:** 3 good
**Contribution:** 3 good
**Rating:** 6
**Confidence:** 2

**Summary:**

This work investigated the impact of quantization on the fine-grained critical-category performance
of DETR-based object detectors.
This work formulated the critical performance via the logit-label transformation of the corresponding categories.

**Strengths:**

This work found that both the conventional PTQ and QAT cause disparate quantization effects on such critical performance.
They theoretically linked the disparate quantization effects with the sensitivity of critical objectives to the quantization weight perturbation and the sharpness of the critical loss landscape in the QAT.
This paper proposed the Fisher-aware mixed-precision quantization scheme and Fisher-trace regularization to improve the critical performance of interest.

**Weaknesses:**

-- The performance improvements are marginal, as shown inTab. 3 and Tab. 4.

**Questions:**

See weaknesses.

---

> ### Author Response · Authors · 2023-11-20
> **Response to Reviewer WkhP**
>
> We thank reviewer WkhP for your thorough review and valuable comments. Hopefully, the following responses can address your concerns.
>
> We would like to note that both the Fisher-aware quantization scheme (both overall and critical) and the Fisher-trace regularization on the DETR model are our contributions derived from our theoretical analysis in Sec. 3. When combined, they lead to an improved critical mAP performance over uniform quantization baselines by up to 1.15% on 4-bit DETR-R50 on the COCO dataset, and up to **10.4%** on COCO-Panoptic where the critical performance is hindered more due to the large number of classes (Sec. 5.3).
>
> Also, we would like to argue that the significance of the empirical improvements should not be solely determined by the value itself. Our results of the Fisher-critical quantization scheme are significant in the sense that 1) the improvements are mostly 3x larger than the standard error induced by the training randomness, and 2) the improvement is consistent across different precisions, architectures, and datasets. This verifies the improvements brought by our proposed method are significant and not random. Furthermore, we find the improvement of our method is higher when applied to more complicated architectures. Critical mAP improves by up to 0.5% on DETR-R101, 0.8% on DAB DETR-R50, and 0.4% on Deformable DETR-R50 with Fisher-critical over Fisher-overall, compared to the 0.2% improvement on DETR-R50. This further shows the potential of the proposed method as larger models may require more compression.
>
> Besides the observable improvements brought by our method, we would like to emphasize that this paper is the first to investigate the impact of quantization on the fine-grained critical-category performance of DETR-based object detectors, which is a practical setting yet largely overlooked by the community. As we take an initial step to tackle this challenge in this work, it will motivate future explorations on the fine-grained impacts of other compression methods in the computer vision area and general machine learning research.

---

### Meta-Review · Area_Chair_U4Ga · 2023-12-12

**Metareview:**

In this paper, the authors propose a Fisher-aware quantization scheme for DETR detectors with a focus on critical categories. Specifically, the authors consider certain categories more important than others, and thus the classification on these categories matters. The authors then approximate the measurement of loss landscape sharpness with Fisher information, and further design Fisher-aware mixed-precision quantization and QAT regularization with Fisher trace. Experiments on COCO detection and COCO panoptic segmentation show improvement over the baselines.

While the paper received unanimous decision of borderline acceptance, the quality of the paper in its current form cannot justify publication at ICLR due to several major weaknesses:

1) The position and story of the paper are weird. The paper talks about a rather general method (fisher-aware quantization) which does not rely on any domain/problem-specific intuitions. Yet the paper chooses to tackle and improve a specific problem (detection) instead of treating it as one of the downstream applications. In addition, the authors further limit their architecture to DETR and choose to tackle a rather uncommon setting by focusing only on critical categories. All these choices greatly limit the scope of the paper and weaken its value.

2) The derivation of sharpness-aware measure using Fisher information is claimed as one of the technical contributions, but it is not that novel as well. There is a rich literature on sharpness-aware loss minimization and this work is not too different. The sharpness-aware loss minimization as minimizing the trace of hessian has been similarly discussed in [a]. The paper is weak in expanding detailed discussions and comparison to these works.

3) The paper lacks the awareness of the latest developments from the standpoint of the detection community. The setting of focusing on critical categories, by itself, is also weird. Besides the practical difficulty of defining the actual subset of critical categories in practical detection problems (and I don't fully agree with the autonomous driving example), a straightforward question is why bother going with this particular setting when one could simply treat all the rest categories as "others?" The story delivered in the paper overall seems a bit created for novelties.

Given the above weaknesses, the AC considers the paper not ready for publication at ICLR.

[a] How Does Sharpness-aware Minimization Minimize Sharpness? ICLR 2023.

**Justification For Why Not Higher Score:**

Please refer to the weaknesses mentioned in detailed comments.

**Justification For Why Not Lower Score:**

N/A

---

### Decision · Program_Chairs · 2024-01-16

Reject